# Anthropometric, Physiological, and Psychological Variables That Determine the Elite Pistol Performance of Women

**DOI:** 10.3390/ijerph19031102

**Published:** 2022-01-19

**Authors:** Vahid Sobhani, Mohammadjavad Rostamizadeh, Seyed Morteza Hosseini, Seyed Ebrahim Hashemi, Ignacio Refoyo Román, Daniel Mon-López

**Affiliations:** 1Exercise Physiology Research Center, Life Style Institute, Baqiyatallah University of Medical Sciences, Tehran 1435916471, Iran; sobhani518@gmail.com (V.S.); drhashemi.pmr@gmail.com (S.E.H.); 2Medicine, Quran and Hadith Research Center, Baqiyatallah University of Medical Sciences, Tehran 1435916471, Iran; sm_hosseini@yahoo.com; 3Facultad de Ciencias de la Actividad Física y del Deporte (INEF—Sports Department), Universidad Politecnica de Madrid, 28040 Madrid, Spain; ignacio.refoyo@upm.es (I.R.R.); daniel.mon@upm.es (D.M.-L.)

**Keywords:** shooting, profile states, strength, balance, anxiety, coping skills, motivation

## Abstract

Shooting is a high-precision sport that depends on many factors to achieve high performance levels. The main objective of this study was to analyze the differences in anthropometric, physiological, and psychological variables by sport level in women air-pistol shooters. Fifteen female pistol shooters, including seven elite national shooters of Iran and eight non-elite shooters, participated in this study. Analyzed variables were grouped into three sections: anthropometric, physiological, and psychological. Anthropometric variables included: height, weight, body mass index, length of leg, arm span, and proportions between variables. Physiological tests include resting heart rate, static and dynamic balance, flexibility, and upper body strength. Additionally, psychological questionnaires of SMS-6 sport motivation, TSCI trait sport-confidence and SSCI state sport-confidence, ACSI-28 athletic coping skills, and SAS sport anxiety scale were used. The Shapiro–Wilks test and independent t-test were used to analyze the data. Effect size and test reliability were calculated using Cohen’s *d* and Cronbach’s alpha, respectively. Our results showed that elite shooters have higher values of dynamic balance (Y-test), upper body strength (sit-ups), and intrinsic motivation, and lower resting heart rate than non-elite. However, no differences were found in the anthropometric variables, nor in anxiety or coping skills. We conclude that physiological and psychological workouts should be included in the shooters’ training programs to improve their performance.

## 1. Introduction

Shooting is a high-accuracy [1] sport in which age is not a limiting variable of performance [2]. However, good skills and physical fitness, such as strength, flexibility, endurance, and balance, are needed [3]. Shooters should have good levels of coordination and control their limbs, joints, and muscles during competition to achieve an optimal performance, because this skill is related to the level of physical and mental fitness [4]. In addition, anthropometric factors are usually important in sports, but in shooting performance, there seems to be no specific body type [5].

Regarding anthropometric factors, measuring body mass index (BMI) [6] and examining the body structure of an athlete, such as height (stand and sit), limb length (arm span, leg length) [7], or some proportions such as sitting height divided by standing height and sitting height related to leg length [8], could determine the athletic balance [9] and the potential to achieve greater success. In this line, some authors noted that tall or long-limbed athletes would have more body sway or tremor. This tremor size in the limbs or body sway could be affected by the feet position [10], age, weight, and height and could have a negative effect on balance [5] and shooting performance [11]. Interestingly, heavy shooters have different body-sway patterns than thinner shooters because the center of gravity has a direct effect on body movement and performance [12]. Furthermore, when shooters are compared to athletes from other sports, they seem to be heavier and shorter [5], which could suggest a preferential morphology profile in the Olympic shooting sport. In contrast, other studies concluded that height was not related to balance [7], body sway does not affect pistol shooting performance [13], and there is no relationship between weight and center of gravity [5].

In addition to balance, shooters need several physiological factors to achieve high levels of performance [14], including muscular strength [15,16] and hand–eye coordination [4]. Thus, the strength levels of core muscles, such as abdominal, spinal, or diaphragm, seem to be important by the shooters [17] and could be critical to reduce body sway and achieve high performance levels [18]. Moreover, the strength level of muscles such as finger flexors and the deltoid [6,19] have been related to the movements of the barrel of a weapon [20], which directly affect performance [21,22].

Another important physiological factor is hand–eye coordination. The coordination of the mind, muscles, eyes, and nervous system seems to be very important since it could create an efficient pattern of movements to help the shooter’s performance [23]. Furthermore, hand–eye coordination is an essential aspect related to the cleanness of the triggering and could be a differential factor between the sport levels [20,22,24]. In addition, elite shooters have a lower heart rate than beginners [25]. This heart rate is directly affected by physiological (fitness) and psychological (anxiety) factors in shooters [26]. However, a high heart-rate variability affects the autonomic nervous system, which can change the level of arousal [27] and could make participants feel more relaxed, thus improving their performance [28].

The last topic in the literature that affects shooting performance is the psychological factors. One of the most studied variables is anxiety. Anxiety levels depend on personality [2], the individual athlete’s point of view [26], and used to be higher in individuals than in team sport [29]. Based on the inverted U theory, each athlete has their own optimal arousal level for performance [30]. In this line, contrary to non-experienced athletes, elite athletes increase arousal levels before a match and reduce it during competition [31]. This fact could be related to the control of emotions, stressful situations [26], and the use of coping skills for managing anxiety [29]. Specifically in shooting, high levels of anxiety involve physiological and psychological symptoms [32] which can lead to differences in firing technique and the gun’s barrel movement [33]. Therefore, the improvement in physical fitness and coping strategies [29] could have direct effects on the heart rate and performance [2]. Furthermore, skills such as self-confidence, defined as an athlete’s feelings and thoughts to achieve success [34], are related to coping strategies and help to cope with anxiety and negative thoughts to improve performance [26]. In this line, shooting faster could be related to more self-confidence than shooting later [35]. Conversely, excess self-confidence can have negative effects on performance [29].

Another psychological factor that modifies performance is motivation. Athletes could be motivated by internal or external reasons. These types of motivation are important for coaches in order to avoid lack of interest in training and unwanted behaviors [36], which can affect performance [37]. Thus, the shooting performance could be modified by a coach’s incentives and praises, which are positively related to the satisfaction and competence of the shooters [29]. Lastly, even though there is not too much information about the effect of the types of motivation on shooting, it is known that motivation could determine athletic performance in other sports [38].

Although there is some evidence about some of the factors affecting shooting per-formance, there are not many reports about female pistol shooters, and some topics remain controversial. Additionally, not many studies have tried to define a complete profile (anthropometric, physiological, and psychological) according to the sport level in women’s Olympic pistol shooting. Consequently, the main objective of this study is to analyse the differences in anthropometric, physiological, and psychological variables by sport level in female air pistol shooters.

## 2. Materials and Methods

### 2.1. Participants

Fifteen female shooters participated in the study; seven were elite athletes and eight were non-elite athletes. To ensure the differences between sport levels, elite shooters were those who met three inclusion criteria: (I) have competed in international competitions, (II) are included in the Iran national team and have an official staff, and (III) have achieved at least three times the minimum score of 560–580 in the last year’s national or international competitions. On the other hand, non-elite shooters require: (I) a minimum of 6 months’ shooting experience, (II) no previous work with the national staff, and (III) a competition result of 500–545 during the last year. Moreover, shooters that have a disease, mental health problems, injuries, or that take illegal supplements or fail to complete all the tests were excluded from the study.

After being fully aware of the stages, benefits, and possible risks of the tests, all the participants signed an informed consent form before the data collection. The study was approved by the Baqiyatallah University of Medical Science of Iran Tehran with number: IR.BMSU.REC.1399.324.

### 2.2. Procedure

The selection of the variables was carried out in three steps: (I) an extensive literature review, (II) several interviews with national and international expert coaches and athletes, and (III) an analysis of the responses was made by the researchers in order to select those variables in agreement with the experts.

Variables were selected following the recommendations of the previous literature and grouped into three sections: anthropometric, physiological [39], and psychological [40,41,42]. Once the final variables were selected, the data collection was carried out during October–November 2020 to ensure similar seasonal conditions.

The researcher who made all the measurements was an authorized examiner of the International Society for the Advancement of Kinanthropometry (ISAK level one). To ensure reliability and measurement accuracy, this researcher performed an additional training program prior to data collection. Additionally, two more referees judged the examiner decision, and they confirmed the measurement. The attempts which had no consensus between the referees and the researcher were rejected.

#### 2.2.1. Anthropometric Test

The anthropometric variables were measured in a single session on the shooters’ training shooting ranges during their rest day, following the official criteria of the International Society for the Advancement of Kinanthropometry, by an authorized examiner. The analyzed variables were age (years), performance (point), height (cm), weight (kg) measured by a digital scale model Beurer ps07 (Beurer company, Germany, Uttenweiler), sitting height (cm), i.e., distance from the highest point on the head to the base sitting surface measured by height tape Sohehnle No.6900.00 (Sohehnle-professional, Germany, Galidor), length of the leg (cm), i.e., distance from the umbilicus to the medial malleoli of the ankle, and armspan (cm), i.e., distance from the tip of the middle finger on one hand to the tip of the middle finger on the other hand with arms abducted to 90°; both tests were measured by KDS diameter F10-02DM (Muratec-kds Corp, Japan, Kyoto).

#### 2.2.2. Physiological Test

Regarding the physiological assessment, all tests were taken in a single session on the shooters’ training shooting ranges during their rest day and repeated three times using the best record, except the Stork test, sit-ups, push-ups, and plank, in which only one attempt was registered.

The following test and protocols were done: (A) Y-test (cm): The starting position is standing on one leg at the stance plate with the toes of the foot at the red line and the other leg touching down lightly just behind the plate. The non-stance foot is reached out in the desired direction, pushing the reach indicator as far possible while maintaining balance. Attempts are made with two legs in three directions, and the maximum reach in each direction is recorded. (B) Stork test (s): Standing barefoot and with hands on hips and the heel of one foot lifted off the ground, the maximum possible time to maintain balance is measured with a time limit of one minute. As soon as the person makes a rotation or hands or heels fall off, the time will stop. (C) Sit-and-reach flexibility test (cm): While sitting on the floor with legs fully extended and the bottom of the feet against the sit-and-reach box, the maximum hands forward movement without flexing the knees is measured. (D) Alternate-Hand Wall-Toss Test (repetitions/s): Starting with the desired hand, the athlete throws the ball towards the wall and catches the ball with the opposite hand and vice versa. The shooter must try to perform the maximum number of repetitions in 30 s. (E) Resting heart rate (beats/min): Shooters are measured at rest using a pulse oximeter model Santamedical sm-519br (Santamedical, USA, Tustin). (F) Push-ups (repetitions/min): Beginning from the prone position, athletes try to raise and lower the body until their limit is achieved. (H) Sit-ups (repetitions/min): To perform valid sit-ups, athletes should go up to touch their knees and return to the starting position.

#### 2.2.3. Psychological Test

Regarding the psychological assessment, shooters had a one-week period to complete the psychological questionnaires sent via email. The following standardized and validated questionnaires were used (see Appendix A). The SMS-6 sport motivation test; our reliability was α = 0.73. The TSCI trait sport-confidence and SSCI state sport-confidence inventories; our reliability was α = 0.98 for both questionnaires. The ASCI-28 athletic coping skills inventory; our reliability was α = 0.71. The SAS sport anxiety scale; our reliability was α = 0.86. Lastly, the sport mental toughness (SMTQ) and sport commitment model scale (SCMS) were evaluated too, but due to our reliability (α = 0.41 and α = 0.48, respectively), both tests were excluded from the results. Before starting to fill in the questionnaires, all tests were previously explained to the athletes.

### 2.3. Statistical Analysis

Variables were described by their mean (*M*) and standard deviation (*SD*). The Kolmogorov–Smirnov test and Shapiro–Wilks test were used to check the normality of the data. The independent *t*-test was used to determine the differences between sport levels. Additionally, the effect size (Cohen’s *d*) was calculated with three cut-off points (*d* = 0.2 small, *d* = 0.5 medium, *d* = 0.8 large), with a 95% interval confidence. The reliability of the psychological questionnaires was calculated using Cronbach’s alpha. The significance level was set at *p* < 0.05. SPSS software version 20 was used to analyze the data.

## 3. Results

Table 1 shows the demographic and anthropometric comparisons by sport level. The results showed that elite shooters had higher performance than non-elite shooters (*p* = 0.001; *d* = 2.47). No differences were found in the rest of the variables (*p* > 0.05).

Regarding the physiological variables, the analysis revealed that elite shooters had higher values of the Y-test left-foot back direction (*p* = 0.044; *d* = −1.16) and a higher number of sit-up repetitions (*p* = 0.008; *d* = −1.63), and lower values in the resting heart rate (*p* < 0.001; *d* = 3.04) than non-elite shooters (Table 2). No differences were found in the rest of the physiological variables (*p* > 0.05).

Regarding the psychological variables (Table 3), elite shooters had higher values of intrinsic motivation (*p* = 0.031; *d* = −1.28) than non-elite shooters. No differences were found in the rest of the variables (*p* > 0.05).

## 4. Discussion

This study tried to analyze the impact of anthropometric, physiological, and psychological variables on the shooting performance by sport level. The main results of this study showed significant differences in the Y-test left–-back(cm), sit-up (repetitions/s), resting heart rate (beat/min), and intrinsic motivation between elite and non-elite shooters. Thus, the pistol shooting performance of women could be influenced somehow by dynamic balance, abdominal endurance, resting heart rate, and motivation.

Regarding the anthropometric profile, finding the proper body type in the sport could lead athletes to achieve success [7]. However, our results showed no differences in the anthropometric variables between elite and non-elite shooters. These results are in accordance with the study by Mon et al. [5], who suggested that, although the weight was related to the body sway of the shooters, neither the height nor the weight had a direct effect on performance. Contrary to our previous results, BMI was related to the performance of female pistol shooters [6]. These differences could be due to the shooting levels and the number of participants (15 shooters divided between elite and non-elite groups in our study vs. 23 not divided in any groups), and the BMI values in both studies, because in the study by Mon et al. [5] the BMI data (24.63 kg/cm^2^) were more similar to our non-elite participants (23.62 kg/cm^2^) than to our elite shooters (21.95 kg/cm^2^). Furthermore, it should be noted that the BMI measurement does not consider whether the weight is muscle or fat, so the body and its shape can be very different with a similar BMI. Accordingly, future studies could measure the fat and lean percentages or use somatocards to have more reliable anthropometric measurements [9].

In addition, contrary to other precision sports such as archery, in which the arm span could determine the performance [7], we did not find correlations between anthropometric variables and performance. Although in pistol shooting, it could be thought that the arm length could generate a greater tremor as a consequence of the kinetic chain [21], there seem to be other non-anthropometric factors that would better explain the barrel movements of the pistol, such as strength [19] or balance [13]. Therefore, in contrast to those studies that pointed out that shooters could have some specific anthropometric characteristics [3], such as being heavier and smaller than the athletes of other sports [5], our results would suggest that there is not a specific profile related to performance in women’s pistol shooting and that the anthropometric factors are more efficient in combination with physical fitness [7]. Nonetheless, the knowledge of relative measures to the full body such as arm musculature or weight, or specific strengths related to the arm such as shoulder abduction or hand grip, could provide additional information on variables that could influence performance in shooting events [15].

In agreement with previous studies, our result showed that elite shooters have greater dynamic balance (left-foot back direction) than non-elite shooters, with a large effect size (*d* = −1.16). Like in other sports, most of our shooters are right-handed. In consequence, they preferably take their left foot back, similarly to our test [20]. This fact could be related to the weight distribution during aiming, as the weight is not equally distributed between the two legs, and the sport level could be a critical factor [24]. In addition, a strong postural balance could increase performance [4] by influencing accuracy and stabilization directly and indirectly [12]. Thus, the wideness of the feet position may improve balance [10] by changing the motions of the center of gravity [12], resulting in athletes with less body sway [5], enhanced aim, and, consequently, improving performance [22].

However, we did not found differences in the rest of balance variables (*p* > 0.05). This fact could be related to the importance of the static vs. dynamic balance effects on performance, with the first one being more relevant [4]. In addition, the balance effect on shooting performance seems to be related to the measurement test, being more effective in those tests similar to the shooting position [43]. Consequently, our results would suggest the need to use specific shooting tests to control the balance effect on performance. Moreover, other factors such as the participants’ age or the number of elite participants could explain partially the absence of balance influence on performance [4]. On the other hand, the effect of an intervention program and specific training programs [14] seems to be a determining factor for balance and performance [16]. In this line, the reduction in trainings during the COVID pandemic has matched elite and non-elite athletes [44], which could determine the absence of differences in balance tests.

In terms of hand–eye coordination, keeping excellent pattern movements in mind helps shooters to connect the muscle and nerve system [23], which improves performance. Furthermore, coordination has a direct effect on the trigger pressure [22] and in the barrel’s movement, thus improving performance [21]. However, our result does not show significant differences between the two groups. This fact could be related to the unspecific test that was carried out, suggesting the need for more specific tests in the future.

Regarding strength, elite shooters exhibited stronger abdominal endurance than non-elite shooters, with a large effect size (*d* = −1.63). These higher strength values could contribute to better control and less tremor [6], reducing the movements of the barrel of the pistol and improving performance [21]. In contrast, upper body strength could be unimportant in shooting performance [16]. This difference in the relevance of strength in performance could be related to the use of a gender-mixed sample, which confirms the unequal performance between genders in pistol shooting [45]. Moreover, technical differences such as the number of shots (60 vs. 10 shots), the weapon (caliber vs. pellet), the shooting time, the target dimensions, or even the use of an intervention strength protocol could be reasons for the performance differences [16].

The last physiological performance-related component is heart rate. As expected, non-elite shooters had a greater resting heart rate than elite shooters with a very large effect size (*d* = −3.04). Generally, the body’s fitness is directly related to the resting heart rate. In this line, elite athletes usually have better cardiovascular and respiratory fitness than non-elite. This fact is associated with thickness of the left ventricle and increase in systolic volume, which ultimately reduces the number of heartbeats [46].

Many studies agree that decreasing heart rate improves concentration and has an influence on performance because of the inverse relationship between vibration and shooting performance. Thus, the heart rate and breathing motions are linked to the body vibration [11]. However, in other shooting modalities, such as shooting performed by police officers, heart rate could be unrelated to performance [32]. Moreover, these differences could be related to the use of a national staff in their programs and the number of hours of fitness training by sport level [17], suggesting the need to have more studies on this topic.

Additionally, heart rate is related to psychological aspects such as arousal and anxiety [26]. Psychological factors such as motivation, self-confidence, and anxiety could be critical to achieve optimal levels of performance [29]. Our findings show that intrinsic motivation has a big influence on performance (*d* = −1.28). Undoubtedly, motivation is one of the most important factors in the success and efforts of athletes and can determine the athlete’s maximum growth and performance [37]. Furthermore, intrinsic motivation, which indicates that an athlete does sport for joy or pleasure, has been proven to be important for performance [47]. However, although motivation is necessary, it could be insufficient for success in other endurance sports [38]. In this line, shooting may have specific motivations requirements due to its special characteristics of maximum precision [1].

Athletes often use coping skills to control themselves while competing [29]. Surprisingly, our finding did not show significant differences between the two groups in terms of anxiety, self-confidence, and coping skills. One possible explanation could be that during the COVID-19 pandemic, many competitions, including the Olympics, World Championships, and internal leagues, were canceled [48], and the number of high-pressure situations was drastically reduced during the last year. In this line, it has been reported that athletes achieved high levels of cognitive and physical anxiety and low levels of self-confidence in competition compared to training [33]. Moreover, the COVID-19 pandemic has completely changed the lives of athletes technically and psychologically [49], and although athletes should be able to perform their workouts remotely by modeling them at home in a safe environment [50], some of their training habits changed significantly. Another reason could be that, in order to avoid long periods together between the researchers and the shooters, the questionnaires were sent via email to the athletes. Therefore, the shooters completed the questionnaires without the presence of the researchers, and without pressure or anxiety, as they had one week time to answer it and a follow-up was performed with the shooters who did not respond within the timeline. In consequence, this aspect may be related to the lack of differences by sport level [51]. Additionally, the age and the effect of experience could be related to the absence of psychological differences by level in our study [29].

Although there is not too much literature related to athletic profiling including anthropometry, physiology, and psychology of female pistol shooters and some of our results seem to be consistent with the previous studies, some limitations should be mentioned. The participant number of this study is limited and using a larger community could improve the statistical power, reduce the interval confidences of the effect size, and could confirm our results. Additionally, the use of more diverse tests, especially a specific shooting test, could add relevant information to the scientific literature. Furthermore, we cannot be sure about the effects of the pandemic data collection on our results, as our data were collected under special conditions. Lastly, even though our study could provide a base for future studies, more information is needed, particularly in the field of women’s pistol shooting, specifically, in order to check the effects of special training conditions which could minimize differences by sport level in precision sports.

## 5. Conclusions

Our findings revealed that a variety of factors could influence shooting performance somehow, among which heart rate, abdominal strength, specific dynamic balance, and intrinsic motivation should be highlighted. However, the data collection during the pandemic period could be related to the absence of results in other variables. Accordingly, specific psychological workouts in addition to physical fitness activities could assist the athlete in reaching optimum performance. Thus, the shooting staff should include physical fitness (balance and strength) and mental programs related to motivation to improve the performance of shooters. Additionally, it is the responsibility of coaches to inform athletes of accurate, scientific information and to assist athletes in flourishing and controlling their motivation levels.

## Figures and Tables

**Table 1 ijerph-19-01102-t001:** Demographic and anthropometric differences by sport level.

Variable	Elite		Non-Elite			Effect Size		Level of Significance
	M	SD	M	SD	t	Cohen’s *d*	IC-95%	*p*
Age (years)	26.29	4.11	25.63	7.42	0.209	−0.11	−1.12 to 0.90	0.838
Performance (points)	571	5.74	536	19.12	4.82	2.47	−3.82 to −1.13	0.001 *
Height (cm)	161.00	2.16	165.06	10.19	−1.029	0.55	−0.48 to 1.58	0.305
Weight (kg)	56.94	8.64	64.10	12.30	−1.284	0.67	−0.36 to 1.71	0.221
Body Mass Index (kg/cm^2^)	21.95	3.17	23.62	4.65	−0.801	0.42	−0.60 to 1.44	0.437
Sitted Height (cm)	86.85	1.77	88.00	2.26	−1.075	0.56	−0.46 to 1.50	0.302
Arm Span (cm)	160.00	2.76	162.50	10.70	−0.598	0.32	−0.69 to 1.33	0.560
Leg Length (cm)	96.14	2.79	97.50	5.65	−0.574	0.30	−0.71 to 1.32	0.386
Sitted Height/Height (cm)	0.539	0.009	0.535	0.039	0.299	−0.14	−1.15 to 0.87	0.773
Sitted Height/Leg Length (cm)	0.904	0.034	0.906	0.069	−0.062	0.04	−0.97 to 1.05	0.951

Notes: Elite (*n* = 7) and non-elite (*n* = 8). M = mean; SD = Standard deviation; Cohen’s *d* = effect size; IC-95% = Interval confidence 95%; *p* = significant level, significant differences are marked with *.

**Table 2 ijerph-19-01102-t002:** Physiological characteristics and their differences by sport level.

Variable	Elite		Non-Elite			Effect Size		Level of Significance
	M	SD	M	SD	t	Cohen’s *d*	IC-95%	*p*
Flexibility (cm)	37.14	7.94	39.75	7.49	−0.654	0.33	−0.68 to 1.35	0.527
Stork test (s)	1.00	0.00	0.84	0.28	1.421	−0.80	−1.86 to 0.24	0.171
Y-test Right—Right (cm)	88.57	3.77	87.12	9.20	0.387	−0.20	−1.22 to 0.80	0.705
Y-test Right—Back (cm)	83.57	11.07	75.37	6.30	1.794	−0.91	−1.97 to 0.15	0.096
Y-test Right—Left (cm)	66.85	2.67	73.75	10.93	−1.618	0.86	−0.19 to 1.92	0.130
Y-test Left—Right (cm)	81.57	4.85	76.00	8.50	1.525	−0.80	−1.85 to 0.24	0.151
Y-test Left—Back (cm)	84.28	9.75	73.13	9.44	2.248	−1.16	−2.25 to −0.06	0.044 *
Y-test Left—Left (cm)	83.85	14.28	83.37	15.98	0.061	−0.03	−1.04 to 0.98	0.952
Alternate-Hand Wall-Toss (repetitions/s)	18.85	2.79	19.75	8.37	−0.268	0.14	−0.86 to 1.15	0.793
Plank (min)	1.57	0.41	1.37	0.73	0.664	−0.33	−1.35 to 0.68	0.505
Sit-up (repetitions/s)	43.42	12.08	23.25	12.56	3.158	−1.63	−2.80 to −0.46	0.008 *
Push-up (repetitions/s)	24.28	4.49	17.50	13.15	1.295	−0.69	−1.73 to 0.35	0.218
Resting Heart Rate (Beat/min)	67.28	2.69	86.75	8.63	−5.705	3.04	1.55 to 4.53	<0.001 *

Notes: Elite (*n* = 7) and non-elite (*n* = 8). M = mean; SD = Standard deviation; Cohen’s *d* = effect size; IC-95% = Interval confidence 95%; *p* = significant level, significant differences are marked with *. Y-test: two legs (right–left), three directions (right–back–left).

**Table 3 ijerph-19-01102-t003:** Psychological characteristics and their differences by sport level.

Variable	Elite		Non-Elite			Effect Size		Level of Significance
	M	SD	M	SD	t	Cohen’s *d*	IC-95%	*p*
Sport motivation	107.00	12.81	98.37	21.06	0.939	−0.49	−1.5 to 0.53	0.365
Intrinsic motivation	21.71	1.60	17.37	4.50	2.410	−1.28	−2.39 to −0.17	0.031 *
Trait sport-confidence	91.28	15.15	85.37	28.68	0.487	−0.25	−1.27 to 0.75	0.634
State sport-confidence	92.00	17.25	85.25	27.93	0.552	−0.29	−1.30 to 0.72	0.590
Sports anxiety	43.14	11.56	38.37	10.43	0.840	−0.43	−1.45 to 0.59	0.421
Athletic coping skills	56.57	5.56	56.50	12.05	0.014	0.00	−1.02 to 1.00	0.989

Notes: Elite (*n* = 7) and non-elite (*n* = 8). M = mean; SD = Standard deviation; Cohen’s *d* = effect size; IC-95% = Interval confidence 95%; *p* = significant level, significant differences are marked with *.

## Data Availability

Data is contained within the article or Appendix A.

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
