# Peer review of "Anthropometric, Physiological, and Psychological Variables That Determine the Elite Pistol Performance of Women"

_ijerph, 2022, doi:10.3390/ijerph19031102_

Round 1
Reviewer 1 Report
General comments:
This study on the effect of various anthropometric, physiological and psychological variables on pistol performance in female shooters is well-structured, clearly written and well explained.
There are a few minor English grammar/expression mistakes which the authors should correct.
Specific comments:
- Regarding the physiological variables, both in the Stork test and in the Plank, the authors state that the “time was stopped” as soon as the participants lost position. How can the authors be sure that the precision of the measured time intervals was correctly controlled? Are the authors convinced (and how?) that the (subjective) reflex delay of the person who was “stopping the time” did not affect the accuracy of the time interval? This is of course related to the magnitude of the time interval (see comment no. 2 below). Please include a relevant comment.
- Table 2: No units are provided for the variable “time” (variables “Stork test”, “Alternate-Hand Wall-Toss”, “Plank”, “Sit-up” and “Push-up”). Please correct. Also please provide a comment regarding the expected precision of the measured time intervals, taking also into account comment 1 above.
- The fact that elite-level athletes have a lower resting heart rate should have been expected, as the resting heart rate level is widely accepted as an indicator of fitness. Please include a relevant comment.
- The authors comment “on the importance of static balance vs dynamic balance” in elite shooting performance. What other variables (from the ones studied by the authors) were expected to be significantly related to performance other than the one found by this study? Was this result affected by the limited sample? Please comment.
- Please also provide a comment as to why the “left foot back direction” was shown to be such a determining factor for shooting performance. In other words, in what manner does this variable affect balance in the static position of the shooters?
- No significant relation was found by the authors between the psychological variables and performance, other than motivation. The authors state that they were surprised by this finding and they comment on the effect of the pandemic. However, the questionnaires the participants had “unlimited time” to fill in the questionnaires. Undoubtedly the questionnaire outcome would have been completely different if they were filled in under pressure and especially before competition! Please comment.
- Finally, would it be possible to translate (at least a sample) of the questionnaires that the authors include in the supplementary files?
Author Response
"Please see the attachment."

Reviewer 2 Report
The aim of the study was to evaluate the influence of physiological, anthropometric and psychological variables on the performance of female shooters.
This kind of studies are of great importance and practicality for coaches and trainers.
Here are some contributions:
- There is no reference to the aim of the study in the abstract. Add the objective of the study in the abstract.
- An appropriate introduction to the subject.
- The researcher who performed the anthropometric measurements has what level of ISAK? specify in the text.
- The article should be written in an orderly fashion.Why start the section on procedures by referring to psychology and then finish this section by specifying the measurements in this area? In the introduction, anthropometry is discussed, then physiology and then psychology, whereas in the procedures the order is different. Maintain an order and structure for the whole article, including the discussion.
- "The main results showed significant differences in the physical fitness and psychological parameters between elite 201 and non-elite shooters". Considering that only 23% of the physiological variables and 16.6% of the psychological variables are different, it seems very brave to make such a statement in the discussion.
- Line 207 "Mon, Zakynthinaki and Calero", wouldn't Mon et al. be more accurate? correcting all similarities in the text.
- Line 210, could it not also be due to the error in this measurement? this measurement does not take into account whether the weight is muscle or fat, so that the body and its shape can be very different with a similar BMI. Taking into account that anthropometric measurements have been made by a professional qualified (ISAK), the correct thing to do would have been to measure the fat and lean percentages and a somatocard, not the BMI.
- Line 218, also knowing arm musculature or an estimate of arm weight could provide important information on variables that influence performance in shooting events.
- Line 275, reference 27 does not say that motivation is not critical in endurance sports, it says "that motivation is necessary but not sufficient for successful athletic performance", but it is necessary/important.
- Check the format of the references: some are in capital letters, some contain the full names (26), others contain the surname + initials (30)…
Such studies are very practical and interesting to read, but I believe that in this case, this study does not add much new to the existing literature.
Author Response
"Please see the attachment."

Reviewer 3 Report
The article is well written but the sample used is too specific and too small to be considered of the importance in which the journal is placed.
Some suggestions for improvement could be, for example:
It would be a good idea to include sub-sections in the procedure. Indicate how long it took to complete the survey.
Cite the most significant bibliography in relation to the choice of physical fitness and psychological tests.
A large number of citations not used in the discussion are used in the discussion.
Author Response
"Please see the attachment."

Round 2
Reviewer 2 Report
The contributions made have been taken into account. Congratulations to the authors.
Reviewer 3 Report
Congratulations to the authors for their efforts to improve the article.